# Heart Rate Variability Parameters to Evaluate Autonomic Functions in Healthy Young Subjects during Short-Term "Dry" Immersion

Liudmila Gerasimova-Meigal * , Alexander Meigal , Nadezhda Sireneva, Maria Gerasimova and Anna Sklyarova

Department of Physiology, Pathophysiology, and Histology, Petrozavodsk State University, 33, Lenina Pr., 185910 Petrozavodsk, Russia
* Correspondence: gerasimova@petrsu.ru; Tel.: +7-911-402-9907

**Abstract:** There is a gap in the current knowledge on the immediate mechanisms of cardiovascular regulation in human subjects within short-term exposure to modeled microgravity using "dry" immersion. Aim. The purpose of the study was to evaluate cardiovascular responses in young healthy subjects during a 45 min session with the help of linear and nonlinear heart rate variability and hemodynamics parameters. The research voluntarily enrolled 33 subjects (18 men, 15 women) aged 19–23 years old. Results. The study showed that systolic and diastolic blood pressure was quite stable, some time-domain parameters of heart rate variability (SDNN, RMSSD, pNN50, etc.) and the frequency-domain (TP, HF, LF, but not VLF) have significantly increased within a 45 min "dry" immersion session. Of the non-linear parameters of heart rate variability, only ApEn significantly decreased during the "dry" immersion session. Conclusion. Our results suggest that a short-term 45 min DI session provokes in young healthy subjects neurogenic autonomic reaction based on the baroreceptor reflex. This provides stable hemodynamics in these subjects along the "dry" immersion session.

**Keywords:** dry immersion; microgravity; heart rate variability; blood pressure; nonlinear dynamics; autonomic function





## 1. Introduction

Ground-based models of microgravity (μG) are increasingly used to simulate real-space weightlessness to allow for the studying of gravity-dependent physiological processes [1,2]. The growing interest in such an unusual environment as μG is associated with its use in the field of space physiology and potential application in the field of rehabilitation due to the specific effects of μG on the human body [3].

Several models of ground-based mG are internationally used, for example, head down tilt, supine bed rest, parabolic flights, free-fall machines, "wet" and "dry" immersion (DI), and unilateral lower limb suspension [1,2,4]. DI refers to the condition of immersion in thermally comfortable water without direct contact with water [3]. Compared to other methods, DI is acknowledged as the most efficient method of modeling microgravity under Earth's conditions, because it triggers the μG-related effects several times faster than, for example, the head down tilt or supine bed rest [5]. In addition to its high efficiency, DI does not require costly tools (aircraft, free-fall machine) and therefore is sparing.

Under DI, the subject's body experiences main physical factors of space flight, such as redistribution of the body's extracellular fluid, supportlessness, and hypokinesia [1,3], which translate primarily to skeletomuscular deconditioning, a decrease in muscle tone [6], and modification of hemodynamics and cardiovascular regulation [3].

The vast majority of studies on artificial ground-based μG are conducted in favor of space exploration. Therefore, in most of them, longer-term DI sessions are usually

chosen, typically from 3 days to 2–3 weeks [1,7], which corresponds to the time period of a short-term space flight. Early effects of DI (hours) on human organisms are available from the studies of long-term DI sessions [7,8]. Only a few studies were devoted specifically to the early effects of DI, for example, the study by Genin et al., 1988 [9].

Even less is known about immediate reactions to DI within minutes or tens of minutes. A study of such immediate reactions would be of both basic and practical interest because it would allow for exploring rapid transient processes between normal Earth G and modeled μG and back. Currently, space exploration industry develops, *inter alia*, in the direction of space tourism, where a space flight lasts a mere 40–60 min, and space tourists often appear as people of middle and older age. Therefore, it would be both practically important for suborbital space tourism and theoretically interesting to study "early rapid" physiological reactions of potential space tourists to the conditions of μG.

In our earlier studies, we have shown that after a single short-term (45 min) DI session, blood pressure (BP) and heart rate (HR) modestly, though significantly, decreased in subjects with Parkinson's disease (PD) [10]. In addition, we reported on an increase in both the time- and frequency- domain parameters of heart rate variability (HRV) across the DI session [10], which is indicative of marked neurogenic parasympathetic and sympathetic reactions to DI. Muscle rigidity and tremors also decreased after a 45 min session of DI [11].

Thus, there is a gap in the current knowledge on mechanisms of cardiovascular regulation within the first minutes of exposure to DI conditions in young healthy subjects. Therefore, we aimed to study cardiovascular responses in young healthy subjects in the conditions of modeled μG. To address this problem, we evaluated linear and nonlinear HRV and hemodynamics parameters during a 45 min session of DI in young healthy subjects.

## 2. Results

At baseline conditions (preDI), systolic and diastolic BP were 101–118 and 58–65 mm Hg, respectively, and HR was 60–76 min$^{-1}$ (Figure 1). During the DI session, BP values were quite stable. Nonetheless, a decrease in both systolic and diastolic BP of 4–5 mm Hg was documented. HR tended to slightly decrease (see Figure 1). Results obtained from men and women did not differ between each other, and therefore were unified in one common study group.

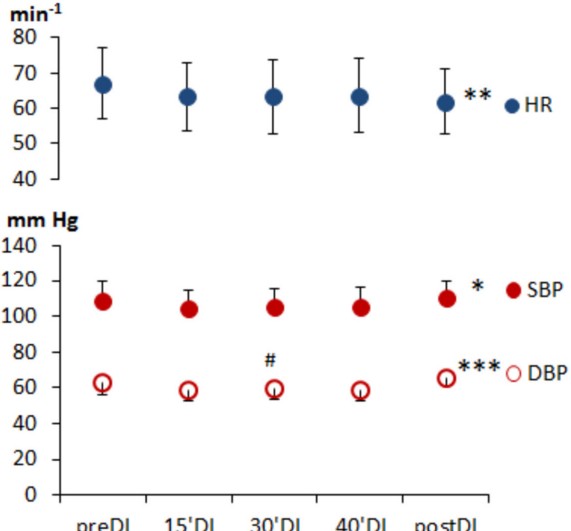

**Figure 1.** Heart rate and blood pressure in subjects during the "dry" immersion session. HR, heart rate; SBP, systolic blood pressure; DBP, diastolic blood pressure. The significance of the Friedman test: * $p < 0.05$, ** $p < 0.01$, and *** $p < 0.001$; the difference from the preDI value # $p < 0.05$.

HRV parameters were highly reproducible among individuals during the DI session. Table 1 presents the results of time-domain HRV analysis during the DI session as Me (25%;

75%). At baseline conditions (preDI), time-domain HRV parameters (minHR, RMSSD, SDNN, pNN50, and TINN) evidenced the high variability of HR, which is the characteristic of healthy young subjects [12,13]. During the DI session, time-domain HRV parameters increased, which indicates the involvement of parasympathetic activity.

**Table 1.** Time-domain results of heart rate variability in subjects during DI session.

| Parameter | preDI | 15′DI | 30′DI | 40′DI | postDI | Significance |
|---|---|---|---|---|---|---|
| RR, ms | 910 (784; 1007) | 984 (903; 1058) | 1002 (835; 1091) | 998 (853; 1071) | 976 (862; 1073) ** | 0.01 |
| HR, $min^{-1}$ | 66 (60; 76) | 61 (57; 67) | 60 (55; 72) *** | 60 (56; 71) *** | 61 (56; 67) *** | 0.001 |
| minHR, $min^{-1}$ | 58 (52; 66) | 54 (49; 61) | 53 (47; 63) * | 53 (47; 65) * | 54 (51; 61) * | 0.01 |
| maxHR, $min^{-1}$ | 76 (71; 86) | 71 (66; 79) | 76 (69; 87) | 79 (68; 90) | 72 (67; 83) | 0.05 |
| SDNN, ms | 43.9 (34.4; 74.6) | 55.4 (42.4; 77.0) | 68.8 (44.7; 87.2) ** | 63.4 (41.9; 86.5) | 53.6 (41.1; 72.6) | 0.01 |
| RMSSD, ms | 56.4 (34.6; 89.7) | 67.9 (45.6; 92.1) | 78.8 (46.0; 118.6) ** | 74.4 (44.6; 110.6) | 58.9 (41.0; 89.4) | 0.01 |
| pNN50, % | 33.43 (8.42; 57.53) | 43.81 (28.29; 60.41) | 46.91 (24.64; 65.03) ** | 44.33 (19.65; 61.15) | 35.89 (19.63; 53.43) | 0.05 |
| TINN | 300.0 (200.0; 413.0) | 305.0 (239.5; 408.0) | 345.5 (237.8; 419.8) | 366.0 (244.8; 413.5) | 304.0 (207.0; 389.0) | n.s. |
| SI | 8.2 (5.1; 10.4) | 7.0 (5.2; 9.2) | 6.3 (4.7; 9.5) * | 6.2 (5.1; 10.2) | 7.8 (5.4; 9.0) | 0.05 |

Study points: before (baseline test—preDI), on the 15, 30, 40 min DI session (15′DI, 30′DI and 40′DI, correspondingly), and 3 min after DI (postDI); the significance is based on the Friedman test with further post-hoc comparisons (the Newman–Keuls test); the difference from the baseline conditions: * $p < 0.05$, ** $p < 0.01$, *** $p < 0.001$, n.s.—non-significant.

The frequency-domain parameters of HRV (TP and its LF and HF bands), presented in Table 2, corresponded with the high variability of HR. That indicates a predominance of the autonomic neurogenic control of HR over humoral-metabolic factors. The ratio of main frequency domains was 56:39:4% (HF > LF >> VLF).

**Table 2.** Frequency-domain results of heart rate variability in subjects during the DI session.

| Parameter | preDI | 15′DI | 30′DI | 40′DI | postDI | Significance |
|---|---|---|---|---|---|---|
| TP, $ms^2$ | 1494 (911; 4982) | 2443 (1461; 4481) | 3219 (1740; 6265) ** | 4003 (1512; 6578) * | 2298 (1441; 4779) | 0.01 |
| HF, $ms^2$ | 876 (394; 2752) | 1668 (906; 2442) | 1542 (793; 3764) * | 1772 (694; 3730) * | 1320 (496; 3165) | 0.01 |
| LF, $ms^2$ | 632 (407; 1606) | 933 (426; 1814) | 1284 (596; 2505) * | 1204 (652; 2072) | 1103 (709; 1751) | 0.05 |
| VLF, $ms^2$ | 50 (24; 105) | 61 (38; 108) | 115 (38; 261) | 65 (41; 147) | 100 (55; 168) | n.s. |
| HF, % | 58.80 (41.07; 66.93) | 57.62 (50.49; 74.92) | 54.09 (37.32; 66.43) | 53.63 (37.40; 67.86) | 50.33 (37.42; 68.93) | n.s. |
| LF, % | 37.81 (30.85; 54.00) | 37.61 (23.90; 46.12) | 40.57 (27.93; 56.71) | 45.69 (27.24; 54.29) | 46.58 (29.30; 54.24) | n.s. |
| VLF, % | 2.88 (1.57; 5.18) | 2.97 (1.56; 4.69) | 3.28 (2.35; 5.14) | 2.63 (1.66; 5.35) | 3.08 (2.27; 7.81) | n.s. |
| LF, n.u. | 39.49 (31.77; 56.79) | 39.35 (24.21; 47.32) | 42.84 (28.87; 61.15) | 46.00 (28.65; 58.81) | 48.05 (29.82; 59.76) | n.s. |
| HF, n.u. | 60.47 (43.19; 68.18) | 60.53 (52.65; 75.79) | 57.12 (38.85; 71.02) | 53.99 (41.17; 71.33) | 51.92 (40.23; 70.16) | n.s. |
| LF/HF | 0.637 (0.466; 1.315) | 0.650 (0.320; 0.899) | 0.840 (0.455; 1.831) | 0.852 (0.402; 1.443) | 0.925 (0.425; 1.487) | n.s. |

For study points and significance, see Table 1.

During the DI session, the marked increase of both time- and frequency-domain HRV parameters (TP and its HF and LF bands) was characteristic of the study points 15′DI, 30′DI, and 40′DI (see Tables 1 and 2, Figure 2). This evidences the autonomic neurogenic parasympathetic and sympathetic response to DI. Still, the structure of the HRV spectrum and LF/HF ratio did not significantly change.

Of the non-linear parameters, presented in Table 3, Poincare plot indices increased during the DI session; the SD2 changed significantly ($p < 0.01$), while SD1 changed non-significantly. The ApEn, as a measure of the HRV regularity and complexity, decreased during the DI session ($p < 0.001$), while SampEn did not change significantly. A lower ApEn may indicate the predominance of the regulatory effect on HR, most likely originating from autonomic neurogenic mechanisms.

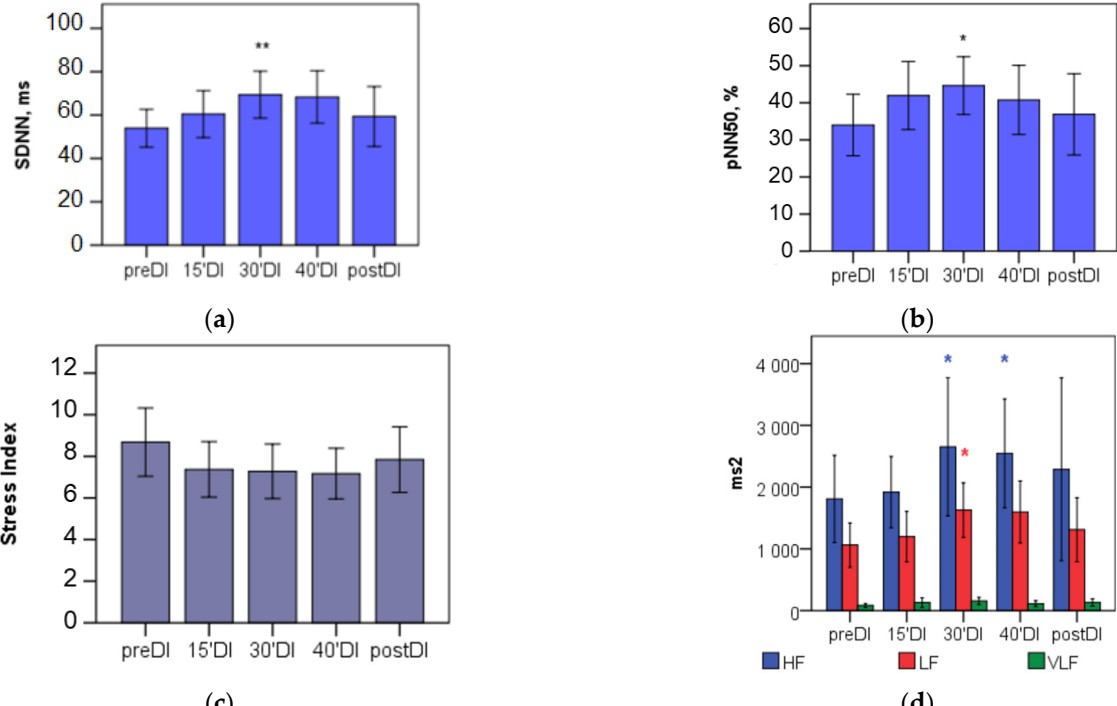

**Figure 2.** Time- and frequency-domain HRV parameters in subjects during "dry" immersion sessions. Horizontal axis depicts study points: before (baseline test—preDI), on the 15, 30, 40 min DI session (15′DI, 30′DI and 40′DI, correspondingly), and 3 min after DI (postDI). (**a**) SDNN, ms; (**b**) pNN50, %; (**c**) Stress index; (**d**) HRV spectrum structure. The difference from the baseline condition: * $p < 0.05$, ** $p < 0.01$.

**Table 3.** Nonlinear results of heart rate variability analysis in subjects during the DI session.

| Parameter | preDI | 15′DI | 30′DI | 40′DI | postDI | Significance |
|---|---|---|---|---|---|---|
| Poincare Plot | | | | | | |
| • SD1, ms | 39.9 (24.5; 63.5) | 48.1 (32.3; 65.2) | 55.9 (32.6; 84.0) | 52.7 (31.6; 78.3) | 41.7 (29.1; 63.3) | n.s. |
| • SD2, ms | 49.2 (40.6; 82.8) | 64.2 (49.3; 82.4) | 75.2 (52.2; 100.1) ** | 72.9 (50.7; 97.6) | 59.8 (47.4; 82.5) | 0.01 |
| • SD2/SD1 | 1.387 (1.186; 1.686) | 1.293 (1.136; 1.591) | 1.379 (1.160; 1.675) | 1.308 (1.222; 1.690) | 1.509 (1.247; 1.760) | n.s. |
| ApEn | 1.127 (1.080; 1.176) | 1.070 (1.012; 1.130) | 1.058 (1.016; 1.162) | 1.082 (1.019; 1.130) ** | 1.069 (1.021; 1.128) ** | 0.001 |
| SampEn | 1.810 (1.589; 1.919) | 1.776 (1.502; 1.892) | 1.754 (1.544; 1.886) | 1.766 (1.570; 1.889) | 1.800 (1.621; 1.948) | n.s. |
| Detrended Fluctutation Analysis (DFA) | | | | | | |
| • Short-term fluctuations, α1 | 0.871 (0.699; 0.929) | 0.758 (0.617; 0.951) | 0.830 (0.646; 0.969) | 0.847 (0.659; 1.028) | 0.848 (0.649; 1.027) | n.s. |
| • Long-term fluctuations, α2 | 0.238 (0.170; 0.388) | 0.254 (0.184; 0.289) | 0.232 (0.183; 0.296) | 0.270 (0.203; 0.300) | 0.269 (0.170; 0.335) | n.s. |

For study points and significance, see Table 1.

In postDI conditions, most of the linear (time- and frequency-domain) and non-linear HRV parameters did not differ from those obtained during DI (see Tables 1–3 and Figure 2). This indicates a high interindividual variability in HRV parameters associated with compensatory autonomic regulation of hemodynamics after recovery from DI and returning to normal conditions. A lower ApEn in postDI conditions indicates the predominance of the regulatory input, which increases the regularity and reduces the complexity of HRV.

## 3. Discussion

The purpose of the study was to evaluate cardiovascular regulation during a short-term DI session in healthy young subjects with the help of linear and nonlinear HRV parameters. We expected a modification of the values of the time- and frequency-domain

parameters of HRV, which would be indicative of baroreceptor responsiveness of the cardiovascular system to the conditions of modeled microgravity. Nonlinear parameters were expected to modify under DI conditions in the direction of higher regularity and lower complexity. In a way, the study can be regarded as a model of a short-term (suborbital) space flight.

The study has shown that some time-domain parameters of HRV (see Table 1) and the power of all bands of the frequency spectrum, except for VLF, significantly increased during a 45 min DI session. Of the non-linear parameters of HRV, only ApEn significantly decreased during the DI session. Marked changes in HRV parameters were characteristic by the 15th minute of DI. By the 30th min of DI, they became statistically significant and kept present until the end of the DI session. As for BP, both systolic and diastolic BP modestly, though statistically significantly, decreased, and HR decreased by an average of 5 bpm during the DI session.

To further understand the physiological mechanisms of such modifications of HRV and hemodynamics, it would be reasonable to discuss the separate and net effect of thermal and immersion conditions, as DI was performed at a water temperature of 32–33 °C. Earlier we have suggested that DI may exert an effect on the cardiovascular system via the following possible physical factors—(1) the "heating" effect, (2) "water immersion" effect, (3) "warm water immersion" effect, and (4) the specific "dry immersion" effect [10].

### 3.1. "Heating" Effect Per Se

Usually, in heat stress experiments, HRV is measured to evaluate autonomic nervous responses during exercises of varied intensity. Still, relevant data on basement data collection conditions, i.e., "pre-exercise" heating conditions, is available from these studies. In young healthy subjects, after 30 min of exposure to moderate heating thermal conditions (exposure to 35 °C air), HR and body temperature significantly increased, the high frequency component (HF%) of HRV significantly decreased, while the low frequency/high frequency (LF/HF) ratio of HRV significantly increased [14]. Still, exposure to 35 °C air within 30 min cannot be regarded as stressful, because in the study by Brenner et al., 1997 [15], heat exposure to air at 40 °C alone did not markedly influence autonomic balance or levels of circulating catecholamines. Thus, a possible moderate heating effect in the present study cannot be regarded as stress and cannot account for observed HRV modification.

### 3.2. "Water Immersion" Effect Per Se

Studies with human immersion in thermoneutral, cold, and hot water have shown that water immersion per se, i.e., during thermoneutral and cold-water immersion, increased stroke volume and decreased heart rate by 15% [16]. During hot water immersion, the heart rate, instead, increased by 32%, which indicated that the temperature effect evidently overrode the immersion effect [16]. During immersion in thermoneutral water (34 °C), an increase in stroke volume led to a 12% increase in cardiac output 10 min after the beginning of water immersion, while HR and BP were not modified [17]. Thus, the effect of water immersion on the cardiovascular system is clearly opposite to that of heating. In addition, a water temperature of 35 °C used during DI sessions must be regarded as rather thermoneutral than heating. This rules out the influence of heating on the results of our study.

### 3.3. "Warm Water Immersion" (WWI) Effect

WWI, often referred to as "passive heat therapy', unlike exposure to warm air, must exert a faster heating effect due to the higher heat conductance of water. In addition, during WWI, immersion acts as a separate independent physical factor. Due to this, WWI is clearly different from heat exposure experiments. Several studies have presented evidence that passive heat therapy is capable of inducing improvements in vascular health in terms of the properties of the arterial wall [18,19]. In the study by Parker et al., 2018 [20], HR and both systolic and diastolic BP decreased significantly in young healthy subjects during

a mere 5 min immersion in the water pool at a temperature of 36 °C. In similar thermal conditions in normal subjects, 15 min WWI caused a significant increase in cardiac and stroke volumes, with no change in HR, central venous pressure, and systemic vascular resistance [21]. Altogether, during short-term WWI, the effect of immersion overrides the heating effect. In addition, water temperatures between 33 and 36 °C, as was previously mentioned, must be regarded as rather thermoneutral than heating [17].

### 3.4. "Dry Immersion" (DI) Effect

Physically, the conditions of DI are very close to those of WWI and, presumably, share common effects. Still, in DI, there is no direct contact with water, which certainly contributes to the subjective perception of DI as a microgravity condition rather than a regular water immersion. In general, the modifications of BP and HRV were unidirectional in those subjects with Parkinson's disease during a single DI session [10]. The data on HRV obtained in the present study correspond with hemodynamic modifications reported from the short-term WWI experiments [14,15].

More specifically, our results suggest that an increase in the power of the TP, HF, and LF bands of frequency spectrum indicates on activation of neurogenic control of HR under DI conditions. As for time-domain parameters, their growth indicates an increase in the parasympathetic (vagal) activity during DI. The sympathetic activity is increased after DI, which is evidenced by an increase in the LF component of the power spectrum and SD2 parameter of the Poincare plot [13]. The sympathetic activity is believed to be associated with the baroreceptor reflex evoked by the "centralization" of circulating blood due to the compression effect of DI on peripheral tissues [1,3,7].

As for the VLF band of the frequency spectrum, which is contributed from humoral-metabolic factors, for example, the renin–angiotensin–aldosterone mechanism [22,23], it was not significantly affected by the conditions of 45 min DI. That might be due to the fact that the subjects were young and healthy. Indeed, in subjects with Parkinson's disease, the VLF band accounted for nearly 50% of the total power of HRV [10] in comparison to 3% in the present study, and that share did not change along the DI session. In addition, the fact that VLF power did not change in our study might be due to the short duration of DI. As for longer-term DI sessions (3–5 days long), the neurogenic component of HRV decreased, which is seen as a decrease in the resting value of the total power of HRV and HF after the dry immersion [24]. At the same time, the sympathetic division of the autonomic nervous system becomes relatively more active after longer-term DI [24,25].

Decreased entropy (ApEn) of HRV informs on the increased regularity, or decreased complexity, of a signal [26]. In our study, ApEn decreased slightly, though significantly. Mäkikallio et al., 1996 [27] reported that ApEn may be dependent on the regularity of breathing patterns. Similarly, Pentillä et al., 2003 [28] found that the complexity (ApEn) of HR behavior can be influenced by vagal blockade changes in breathing patterns. We earlier reported a phenomenon of stronger cardiorespiratory coupling in the study with a deep breathing test [29]. However, we did not measure the respiratory rate in this study.

In conclusion, our results suggest that a short-term 45 min DI session provokes in young healthy subjects neurogenic autonomic reactions based on the baroreceptor reflex. This provides for stable hemodynamics in these subjects along the DI session. We assume that these data would be helpful for the application of DI programs for monitoring immediate physiological reactions in healthy subjects either during/before a short-term space flight. In addition, DI is supposed to be eventually introduced into rehabilitation practices, e.g., in subjects with muscle tone disorders (parkinsonism or cerebral palsy) or with arterial hypertension. In that respect, it would be essential to know all possible reactions to the condition of DI, either in younger or older subjects.

We figure that the present study can be expanded in the direction of a more detailed, e.g., minute-by-minute, monitoring and examination of cardiovascular reactions to the condition of DI, and also call for the study of cerebral hemodynamics during a short-tern session of DI and cardiorespiratory coupling.

## 4. Materials and Methods

### 4.1. Participants

The study group was formed according to the criteria for the safe use of DI as well as for the analysis of HRV, which were earlier published [10]. Participants were recruited to the study by invitation, in which the purpose of the study and applied methods were presented. Then, participants who accepted the invitation were screened for exclusion criteria. In addition, some basic examinations (body temperature and BP, HR, body weight, height) were conducted. Finally, enrolled participants were screened with electrocardiography for possible arrhythmia. The participants were selected on the principle of the uniformity of their BMI (to exclude those overweight) and age. In addition, approximately equal rates of females and males were chosen. Then, there are some exclusion criteria for taking a session of DI [11]. Among these are chronic disease (e.g., arterial either hyper- or hypotension, thrombophlebitis), acute inflammatory diseases, brain trauma in anamnesis, cardiac arrhythmia.

The study enrolled 33 apparently healthy individuals aged 19–23 years old without chronic somatic and neurological diseases. The anthropometric characteristics of participants are presented in Table 4.

**Table 4.** Anthropometric characteristics of the participants at the time of their inclusion in the study.

| Parameter | Men (*n* = 18) | Women (*n* = 15) |
|---|---|---|
| Body Mass, kg | 73.9 ± 7.3 | 60.8 ± 7.2 |
| Height, m | 1.84 ± 0.05 | 1.66 ± 0.05 |
| BMI [1] | 21.9 ± 2.0 | 22.0 ± 2.4 |

[1] BMI, body mass index.

In anamnesis, none of the subjects had brain trauma, including those associated with sport activity, for example, boxing and football. The study included individuals who did not take drugs with a noticeable effect on autonomic regulation and/or heart function, for example, adrenal β-blockers. Cardiac arrhythmias was a major non-inclusion criterion for perfect HRV analysis. A comprehensive verbal explanation was provided to all participants, and written informed consent was obtained from all participants. The study protocol was approved by the joint Ethics committee of the Ministry of Healthcare of the Republic of Karelia and Petrozavodsk State University (Statement of approval No. 31, 18 December 2014). Prior to the DI session, an active orthostatic test for orthostatic tolerance was performed on all subjects. None of the subjects experienced orthostatic hypotension [10,29].

### 4.2. The DI Session

The condition of DI was induced with help of the "Medical Facility of Artificial Weightlessness" (MEDSIM, Center for Aerospace Medicine and Technologies, State Scientific Center of Russian Federation "Institute of Biomedical Problems", Moscow, Russia) which is housed in Petrozavodsk State University. The procedure of DI was described in detail in our earlier papers [10,11,29,30]. MEDSIM looks like a bathtub filled with 2 m$^3$ of fresh water. This bathtub was covered with a thin waterproof film of a large size, which allows you to wrap the subject's body. MEDSIM was equipped with a metal engine-driven platform, with the help of which the subject was immersed in water. At the starting point, the platform was positioned above the water level, which allowed the subject to lie supine on the platform. When the DI procedure begins, the platform is lowered into the water so that the subject remains immersed in the bath, wrapped in a film, with the head and upper chest "out of water". For more information on the physics and procedure of DI, see [7,11].

The water temperature in the bathtub was set at 32 °C. Before the procedure, subjects visited the toilet, as DI has a strong diuretic effect, and drank 200 mL of fresh water. Then the subjects were given the opportunity to familiarize themselves with the experimental conditions for 10–15 min, lying supine on a platform wrapped in a cotton sheet for comfort.

Blood pressure was measured at the 10th minute (UA-767, A&D Company Ltd., Tokyo, Japan). If the blood pressure was not higher than 140/80 mm Hg, the DI procedure was approved to begin. Subjects were immersed in water for 45 min, with the possibility of stopping the procedure on demand.

### 4.3. Outcome Measures

Data were collected in the following study points: before (baseline test—preDI), at the 15, 30, 40 min point of the DI session (15′DI, 30′DI and 40′DI, correspondingly), and 3 min after DI (postDI).

ECG in standard lead II was recorded for 5 min using a "VNS-Spektr" device (Neurosoft Ltd., Ivanovo, Russia). ECG records were visually checked for stationarity, and all artifacts were manually corrected. For the analysis of HRV, only ECG recordings without arrhythmias on the ECG were taken into account. Further analysis was conducted according to international standards for measurement, physiological interpretation, and clinical use of HRV [12,13]. Linear and non-linear parameters were computed using Kubios Standard version 3.5.0 software (University of Eastern Finland, Kuopio, Finland; [31]).

Time-domain HRV analysis was comprised of assessment of the HR, MeanRR, HRmin, HRmax, standard deviation (SDNN), root mean squared difference (RMSSD), the proportion of successive intervals >50 ms (pNN50, %) of normal RRi (NN), and triangular interpolation of the RR histogram index (TINN).

Frequency-domain HRV analysis was comprised of an assessment of the total power (TP) spectrum of RRi, power spectrum at high- (HF; 0.15–0.40 Hz), low0 (LF; 0.04–0.15 Hz), and very-low-frequency bands (VLF; <0.04 Hz), spectrum structure (% VLF, % LF, % HF, LF n.u, HF n.u.), and the LF/HF ratio.

The analysis of non-linear HRV parameters was comprised of an assessment of the indices of the Poincaré ellipse (SD1 and SD2), sample (SampEn) and approximate (ApEn) entropy, and parameters of detrended fluctuation analysis (DFA), with self-similarity indices for short ($\alpha$1) and long time intervals ($\alpha$2) [13,31,32].

The studied HRV parameters have the following physiological value [13]:

- Time-domain parameters
  - The SDNN and RMSSD inform on the variation of parasympathetically-mediated respiratory sinus arrhythmia.
  - The pNN50 is closely correlated with parasympathetic nervous activity.
  - The TINN and RMSSD can jointly distinguish between normal heart rhythms and arrhythmias.

- Frequency-domain parameters
  - HF power reflects the parasympathetic activity and is related to the respiratory cycle.
  - LF power is frequently considered to be related to sympathetic nervous activity. However, the LF power may be produced by both the PNS and SNS, and BP regulation via baroreceptors.
  - VLF power correlates with renin–angiotensin and endothelial influences on the heart.
  - LF/HF ratio measures "sympatho-vagal balance", but this model is challenged.

- Nonlinear parameters
  - ApEn values indicate the predictability of fluctuations in successive RR intervals.
  - DFA describes brief ($\alpha$1) and long-term ($\alpha$2) fluctuations. $\alpha$1 reflects the baroreceptor reflex, while $\alpha$2 reflects the regulatory mechanisms that limit fluctuation of the RR interval.
  - the non-linear metric SD1 of Poincare plot is identical to the RMSSD
  - The SD2 measures correlate with LF power and baroreflex sensitivity.
  - SD1/SD2 correlates with the LF/HF ratio.

Systolic and diastolic BP and HR were measured at the end of the ECG recording using a digital tonometer UA-705 (A&D Company Ltd., Tokyo, Japan).

*4.4. Statistical Analysis*

Data were analyzed using IBM SPSS Statistics 21.0 software (SPSS, IBM Company, Chicago, IL, USA). Within the DI session, ANOVA and a non-parametric Kruskal–Wallace test were used to compare HRV and hemodynamic parameters between study points. If data were available on subjects from pre-DI, 30′, 40′ DI, and postDI ($n$ = 17), the Friedman SPSS test, followed by post-hoc comparisons (Newman–Keuls test), was used to evaluate differences between HRV and hemodynamic parameters. The significance was considered at $p < 0.05$.

**Author Contributions:** Conceptualization, L.G.-M. and A.M.; methodology, L.G.-M. and A.M.; validation, L.G.-M. and A.M.; formal analysis, L.G.-M., M.G., A.S. and N.S.; investigation, L.G.-M., M.G., A.S. and N.S.; resources, A.M.; data curation, L.G.-M., M.G., A.S. and N.S.; writing—original draft preparation, L.G.-M. and A.M.; writing—review and editing, L.G.-M. and A.M.; visualization, L.G.-M.; supervision, L.G.-M.; project administration, A.M.; funding acquisition, A.M. All authors have read and agreed to the published version of the manuscript.

**Funding:** This research was funded by the Ministry of Science and Higher Education of the Russian Federation (Theme No. 0752-2020-0007).

**Institutional Review Board Statement:** The study was conducted in accordance with the Declaration of Helsinki, and approved by the joint Ethics committee of the Ministry of Health care of the Republic of Karelia and Petrozavodsk State University (Statement of approval No. 31, 18 December 2014).

**Informed Consent Statement:** Informed consent was obtained from all subjects involved in the study.

**Data Availability Statement:** Not applicable.

**Acknowledgments:** The authors thank the subjects for their participation.

**Conflicts of Interest:** The authors declare no conflict of interest.

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
