# Peer review of "Heart Rate Variability Parameters to Evaluate Autonomic Functions in Healthy Young Subjects during Short-Term “Dry” Immersion"

_physiologia, doi:10.3390/physiologia3010010_

Round 1

Reviewer 1 Report

The article investigates cardiovascular responses in young healthy subjects during a 45-min session with help of linear and nonlinear HRV and hemodynamics parameters. This work has a potential and my comments are as follows:

- The Abstract should contain answers to the following questions: What

problem was studied and why is it important? What methods were used? What are the important results? What conclusions can be drawn from the results? What is the novelty of the work and where does it go beyond previous efforts in the literature? Add the main findings and objective of the current study in the abstract.

-The Introduction should make a compelling case for why the study is useful

along with a clear statement of its novelty or originality by providing relevant information and providing answers to basic questions such as: What is already known in the open literature? What is missing (i.e., research gaps)? What needs to be done, why and how? Clear statements of the novelty of the work should also appear briefly in the Abstract and Conclusions sections.

-Enhance result discussion section by including more physical justifications

in the revised draft.

- For enhancing the introduction section with the new publications, old references may be replaced with new ones such as:

On behavioral response of ciliated cervical canal on the development of electroosmotic forces in spermatic fluid

Dynamism of a hybrid Casson nanofluid with laser radiation and chemical reaction through sinusoidal channels

Author Response

We appreciate very much all comments of the Reviewer.

We have revised the Abstract. More specifically, we outlined subheadings and avoided acronyms.

We put the purpose of the study more straightforward in the Introduction section.

Also, we have added the Discussion section with new sentences on practical use, novelty and perspectives of the study.

In addition, we have substantially re-written the methods section. We put more attention to details of the dry immersion session.

As for the recommended studies, we cannot refer to them because they are not in the field of our study.

Reviewer 2 Report

DEAR AUTHORS

Manuscript entitled HRV Parameters to evaluate Autonomic Function in Healthy Young Subjects during Short-Term Dry Immersion

GENERAL COMMENTS

Clarity of content and adequacy to scientific language was demonstrated throughout the manuscript. The article is interesting as a whole; however, some adjustments are needed.

In order to facilitate the readers' understanding, I suggest the following changes:

I suggest that the authors substantially improve the research rationale

Why should this method be used?

What is the clinical applicability of the findings of this study?

Statistical analysis

I suggest that the authors add effect size

Author Response

Q1: I suggest that the authors substantially improve the research rationale.

Q2: Why should this method be used?

Answer:

We appreciate very much all comments of the Reviewer.

We substantially modified the third last paragraph of the Results section. We stressed the significance of immediate reactions to artificial microgravity. Also, we added a sentence on practical rationale of the study. Currently, this paragraph is as the following:

Even less is known about immediate reactions to DI within minutes or tens of minutes. A study of such immediate reactions would be of both basic and practical interest because it allowed exploring rapid transient processes between normal Earth G and modeled μG, and back. Currently, space exploration industry develops, inter alia, in the direction of space tourism, where a space flight lasts mere 40-60 minutes, and space tourists often appear as people of middle and older age. Therefore, it would be both practically important for suborbital space tourism and theoretically interesting to study “early rapid” physiological reactions of potential space tourists to the conditions of μG.

Q3: What is the clinical applicability of the findings of this study?

Answer:

We completed the last sentence in the Discussion section with that information. Currently, this sentence is as the following:

We assume that these data would be helpful for application of DI programs for monitoring immediate physiological reactions either in healthy subjects during/before a short-term space flight. In addition, DI is supposed to be eventually introduced to rehabilitation practices, e.g. in subjects with muscle tone disorders (parkinsonism or cerebral palsy) or with arterial hypertension. In that respect, it would essential to know possible reactions to the condition of DI, either in younger or older subjects.

Q4: I suggest that the authors add effect size.

Answer:

Effect size (Cohen’s d-statistics) refers to formula M1-M2/pooled SD, which helps to evaluate difference between two non-paired datasets. However, in our study, multiply (n>2) comparisons were conducted with help of Friedman’s test. Therefore, Cohen’s statistics is not applicable in our study. We regard Friedman’s tests as rather rigor and relevant.

Reviewer 3 Report

This manuscript analyzed the effects of Dry Immersion (water immersion without contact between the subject and the water) in HRV (I suppose Heart Rate Variability, as no definition is provided). The authors must check the template of the journal in order to accomplish the section order and  the required information (type of study). The authors also need to provide extensive information about the procedure used.

-Indicate the type of paper in the line over the title

-Avoid using non-defined acronyms in the title and abstract, p.e.:  HRV

-According to this journal, Materials and Methods section must be section number 2, the one before Results section and not the last after Discussion.

            -How were participants recruited?

            -Specify selection criteria according to CONSORT.

-Line 230: Please used plural: “Anthropometric characteristics of participants are presented”

-Line 249: the procedure of DI must be explained in this paper, as it must by itself contain enough information to be reproduced without additional papers.

Author Response

We appreciate very much all comments of the Reviewer.

Q1: Indicate the type of paper in the line over the title.

Answer: The type of paper is indicated - Article (original research manuscript).

Q2: Avoid using non-defined acronyms in the title (HRV).

Answer: HRV is substituted with Heart Rate Variability in the title.

Q3: How were participants recruited?

Participants were recruited to the study by the invitation, in which the purpose of the study and applied methods were presented. Then, participants who accepted the invitation were screened for exclusion criteria. In addition, some basic examinations (body temperature and blood pressure, heart rate, body weight, height) were conducted. Finally, enrolled participants were screened with electrocardiography for possible arrhythmia.

Q4: Specify selection criteria according to CONSORT.

The participants were selected on the principle of uniformity of their BMI (to exclude overweight) and age. Also, approximately equal rate of females and males was chosen. Then, there are some exclusion criteria for taking a session of dry immersion (Miroshnichenko et al., 2018). Among these, chronic disease (e.g. arterial either hyper- or hypotension, thrombophlebitis), acute inflammatory diseases, brain trauma in anamnesis, cardiac arrhythmia.

Q5: Line 230. Please us plural “Anthropometric characteristics of participants are presented”

Answer: Put in accordance.

Q6: Line 249. The procedure of DI must be explained in this paper, as it must by itself contain enough information to be reproduced without additional papers.

Answer: We added many details to description of the DI procedure to the Methods section. Currently, this paragraph is as the following:

The condition of DI was induced with help of the “Medical Facility of Artificial Weightlessness” (MEDSIM, Center for Aerospace Medicine and Technologies, State Scientific Center of Russian Federation “Institute of Biomedical Problems,” Moscow, Russia). The procedure of DI is in-detail described in our earlier papers [10,11,29,30]. The duration of the DI session was 45-min (short-term DI). During DI, subjects were lying supine, head-out-of-water, in a bathtub with thermally comfortable water (32–33°C). MEDSIM looks like a bathtub filled with 2 m3 of fresh water. This bathtub was covered with a thin waterproof film of large size, which allows you to wrap the subject's body. An engine-driven lifting platform was installed on the floor of the bath. At the starting point, the platform was located above the water level, which allowed the subject to lie supine on the platform. When the DI procedure begins, the platform is lowered into the water so that the subject remains immersed in the bath, wrapped in a film, with the head and upper chest “out of water”. For more information on the physics and procedure of DI, see [7,11].

The water temperature in the bathtub was set at 32°C. Before the procedure, subjects visited the toilet, as DI has a strong diuretic effect, and drank 200 ml of fresh water. Then the subjects were given the opportunity to familiarize with the experimental conditions for 10-15 minutes, lying supine on a platform wrapped in a cotton sheet for comfort. Blood pressure was measured at the 10th minute (UA-767, A&D Company Ltd., Japan). If the blood pressure was not higher than 140/80 mm Hg, the DI procedure was approved to begin. Subjects were immersed in water for 45 minutes with the possibility of stopping the procedure on demand.

Reviewer 4 Report

This study aimed at studying cardiovascular responses in young, healthy subjects in the 64 conditions of modelled μG, evaluated with linear and nonlinear HRV and hemodynamics parameters during a 45-min session of DI in healthy young 66 subjects. 

The introduction section is well-written. 

Regarding the methods section, I would suggest clarifying the inclusion and exclusion criteria and adding information about where the experiment took place. Moreover, I would suggest describing in detail the experimental session. The authors only mention in the manuscript that "During DI, subjects were lying 250 supine, head-out-of-water, in a bathtub with thermally comfortable water (32–33°C)."

Generally, the Methods section should be rewritten. 

Also, please provide the physiological value of each HRV index you have included in your manuscript. 

Please conduct a sample size analysis. 

I would suggest describing the practical value of your study and adding a paragraph with perspectives.

Author Response

We appreciate very much all comments of the Reviewer.

Q1. Regarding the methods section, I would suggest clarifying the inclusion and exclusion criteria and adding information where the experiment took place. Moreover, I would suggest describing in detail the experimental session.

We have re-written sections 4.1 and 4.2 substantially.

Q2. Provide the physiological value of each of HRV index you have included in your manuscript.

Answer:

The studied HRV parameters have the following physiological value (Shaffer):

  • Time-domain parameters
    • The SDNN and RMSSD inform on the variation of parasympathetically-mediated respiratory sinus arrhythmia
    • The pNN50 is closely correlated with parasympathetic nervous activity
    • The TINN and RMSSD can jointly distinguish between normal heart rhythms and arrhythmias
  • Frequency-domain parameters
    • The HF power reflects the parasympathetic activity and is related to the respiratory cycle
    • The LF power is frequently considered to be related to the sympathetic nervous activity. However, the LF power may be produced by both the PNS and SNS, and BP regulation via baroreceptors
    • VLF power correlate with renin–angiotensin and endothelial influences on the heart
    • LF/HF ratio measures “sympatho-vagal balance”, but this model is challenged.
  • Nonlinear parameters
    • ApEn values indicate predictability of fluctuations in successive RR intervals.
    • DFA describes brief (α1) and long-term (α2) fluctuations. α1 reflects the baroreceptor reflex, while α2 - the regulatory mechanisms that limit fluctuation of the RR interval.
    • the non-linear metric SD1 of Poincare plot is identical to the RMSSD
    • The SD2 measures correlates with LF power and baroreflex sensitivity.
    • SD1/SD2 is correlated with the LF/HF ratio

Q3. Please conduct a sample size analysis.

Answer:

Effect size (Cohen’s d-statistics) refers to formula M1-M2/pooled SD, which helps to evaluate difference between two non-paired datasets. However, in our study, multiply (n>2) comparisons were conducted with help of Friedman’s test. Therefore, Cohen’s statistics is not applicable in our study. We regard Friedman’s tests as rather rigor and relevant.

Q4. I would suggest describing the practical value of your study and adding a paragraph with perspectives.

We completed the last sentence in the Discussion section with that information. Currently, this sentence is as the following:

We assume that these data would be helpful for application of DI programs for monitoring immediate physiological reactions either in healthy subjects during/before a short-term space flight. In addition, DI is supposed to be eventually introduced to rehabilitation practices, e.g. in subjects with muscle tone disorders (parkinsonism or cerebral palsy) or with arterial hypertension. In that respect, it would essential to know all possible reactions to the condition of DI, either in younger or older subjects.

In addition, we added two sentences on the perspectives (the last two sentences in the Discussion section):

We figure out that the present study can be expanded in the direction of a more detailed, e.g. minute-by-minute, monitoring and examination of cardiovascular reactions to the condition of DI. Also, we consider of studying cerebral hemodynamics during a short-tern session of DI and cardiorespiratory coupling.

Round 2

Reviewer 1 Report

Authors have done the required amendments and article is ready for publication.

Author Response

Dear Reviewer, thank you for your valuable comments.

Reviewer 2 Report

Dear Authors

I suggest that the authors improve the rationale

 of the research.

Author Response

(The authors gave the same response as above.)

Reviewer 3 Report

Thank you for accepting my suggestions

Author Response

(The authors gave the same response as above.)

Reviewer 4 Report

Thank you for adequately addressing all the comments. However, I have one more comment. I suggested conducting a sample size analysis to evaluate if the number of participants was enough, and this reflects the power of the study to conclude

Author Response

Dear Reviewer, thank you for your valuable comments.

Comment: Thank you for adequately addressing all the comments. However, I have one more comment. I suggested conducting a sample size analysis to evaluate if the number of participants was enough, and this reflects the power of the study to conclude.

Answer:

We agree that the issues of power and sample size are extremely important. We considered these issues.

First, power analysis is applied to the studies, which are aimed to confirm/reject some kind of intervention/treatment. In these studies, size effect is, indeed, is essential to make conclusions on the outcome. Usually, these studies represent clinical trials. Therefore, such kind of studies mostly refer to rather “clinical relevance” than “statistical significance”. Second, clinical trials usually preceded by so-called pilot studies, which help to spot the effect and to preliminarily evaluate its significance. Namely, at that stage power analysis is needed to estimate sample size (how many subjects/cases would reveal the effect) (Power and Size Analysis, by L. Douglas Case and Walter T. Ambrosius, p. 377, in Topics in Biostatistics, ed. Walter Ambrosius, Human Press, 2007) according to formula 1 (p. 397, Sample size determination).

We suppose that this approach is not applicable because our study is designed as a repetitive measurement, not just a pre-post comparison. It includes 5 study points (pre-DI, 15th, 30th, 40th min and post-DI). Also, the HRV data are variable by its nature (SD is originally high). Therefore, Friedman’s test for repetitive comparisons was accounted as the most relevant to treat the data. Power of non-parametric Friedman’s test is comparable to that of ANOVA (Donald et al., 1993). Friedman’s test is rather strict as it considers Newman-Keuls post-hoc test (p is subdivided by the number of study points).

Donald W. Zimmerman & Bruno D. Zumbo (1993) Relative Power of the Wilcoxon Test, the Friedman Test, and Repeated-Measures ANOVA on Ranks, The Journal of Experimental Education, 62:1, 75-86, DOI: 10.1080/00220973.1993.9943832

As our study cannot be regarded as a clinical trial, but rather as a pilot physiological study to trace the effect of Dry Immersion on HRV we were not able to predict the effect of DI on each of the HRV parameters. Therefore, it was not possible to compute the sample size by the formula 1 (see above).

In cases of physiological pilot studies, even 10 subjects allow spotting the effects, and 30 subjects looks as appropriate minimum for appropriate. This assumption is based on “INTRODUCTION TO POWER ANALYSIS” (https://stats.oarc.ucla.edu/other/mult-pkg/seminars/intro-power/).

Altogether, we regards that the number of subjects in our pilot study is likely relevant to spot the effect. This study can serve as good basement for further clinical study.

Round 3

Reviewer 4 Report

I believe that the authors have adequately revised the manuscript. I suggest publication.